# A unifying framework for interpreting and predicting mutualistic systems

Feilun Wu[1], Allison J. Lopatkin[1], Daniel A. Needs[1], Charlotte T. Lee [2], Sayan Mukherjee[3] & Lingchong You[1,4,5]

Coarse-grained rules are widely used in chemistry, physics and engineering. In biology, however, such rules are less common and under-appreciated. This gap can be attributed to the difficulty in establishing general rules to encompass the immense diversity and complexity of biological systems. Furthermore, even when a rule is established, it is often challenging to map it to mechanistic details and to quantify these details. Here we report a framework that addresses these challenges for mutualistic systems. We first deduce a general rule that predicts the various outcomes of mutualistic systems, including coexistence and productivity. We further develop a standardized machine-learning-based calibration procedure to use the rule without the need to fully elucidate or characterize their mechanistic underpinnings. Our approach consistently provides explanatory and predictive power with various simulated and experimental mutualistic systems. Our strategy can pave the way for establishing and implementing other simple rules for biological systems.

---

[1] Department of Biomedical Engineering, Duke University, Durham, NC 27708, USA. [2] Department of Biology, Duke University, Durham, NC 27708, USA. [3] Departments of Statistical Science, Mathematics, Computer Science, and Bioinformatics & Biostatistics, Duke University, Durham, NC 27708, USA. [4] Center for Genomic and Computational Biology, Duke University, Durham, NC 27708, USA. [5] Department of Molecular Genetics and Microbiology, Duke University School of Medicine, Durham, NC 27710, USA. Correspondence and requests for materials should be addressed to L.Y. (email: you@duke.edu)

**M**utualism, where two or more populations provide reciprocal benefit, is an essential type of ecological interaction[1]. In marine ecosystems, coral reefs are based on mutualistic interactions between coral and algae, and provide ecosystem services for humans and habitats for diverse organisms[2]. Plant–bacterial mutualism is estimated to generate 60% of the annual terrestrial nitrogen input[3]. Mutualism also influences microbial community structures and is the cornerstone of various microbial metabolic tasks[4,5]. Although mutualistic coexistence is beneficial in maintaining the biodiversity, function and stability of ecosystems, under some conditions mutualistic systems can collapse, where one or more mutualistic partners is lost, and the persisting partners also experience a reduction in fitness. This perturbation can further trigger the loss or invasion of other populations and alter ecosystem functions[6–9]. A framework to interpret and predict mutualistic outcomes is useful to prevent undesirable system behaviors and provide guidance for modulating and engineering mutualistic systems.

Quantitative rules have been developed to elevate our understanding and provide predictive power for various biological systems[10–13]. However, such a framework is not yet available for determining mutualism outcomes. Main barriers in developing such a framework are the diversity of mutualistic interaction mechanisms and the complexity of underlying dynamics. Indeed, even engineered mutualistic systems that are by-design capable of cooperation, may not coexist. For example, it is still difficult to predict a priori whether an engineered microbial auxotrophic pair can persist or not[14–16]. Previously, theoretical criteria in the form of inequalities have been developed for specific mutualistic

systems such as cross-feeding mutualisms[17], plant–pollinator mutualisms[18], seed-dispersal mutualisms[19], ant–plant mutualisms[20], and plant–mycorrhizal mutualisms[21]. These criteria depend on the underlying mechanisms *assumed* in the models and are not applicable to other types of mutualistic systems. General criteria have been developed[22–25], such as the classic criterion which states that intraspecific competition must be greater than mutual benefit for a mutualistic system to be stable[26]. However, these usually describe transitions between stable coexistence and unbounded growth, and fail to address the transitions between coexistence and collapse and other mutualism dynamics[27,28] (Supplementary Figure 1, Supplementary Note 1).

Here, we establish a general framework for predicting and interpreting mutualistic systems. We first generate a wide variety of mutualism mathematical models and identify a general rule that predicts mutualism outcomes for all these models. We then develop a calibration procedure using support vector machines (SVMs) to apply the rule to various simulated and experimental systems with different layers of complexities. The interpretation and predictability provided by our framework demonstrate the feasibility of describing a class of diverse biological systems with a simple quantitative rule.

## Results

**Abstraction reveals a general rule.** To reveal any commonality of mutualistic systems, we first summarized the logic of mutualism (Fig. 1a). Mutualism can be defined as the collective action of two or more populations, where each population produces benefit ($\beta$) that reduces the other's stress ($\delta$) at a cost ($\varepsilon$) to itself. $\beta$, $\delta$, and $\varepsilon$

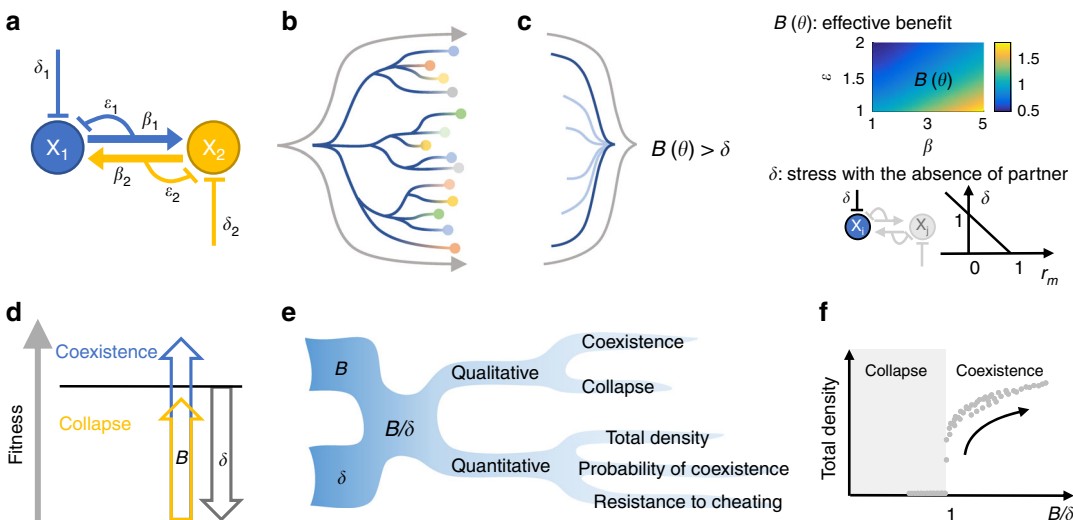

**Fig. 1** $B$ and $\delta$ are two driving forces that determine qualitative and quantitative mutualistic outcomes. **a** The basic logic of mutualistic systems. The two partner populations are denoted by $X_1$ and $X_2$. $\beta_1$ and $\beta_2$ describe the level of benefit. $\varepsilon_1$ and $\varepsilon_2$ describe the cooperation cost of providing benefit. The two populations also experience stress $\delta_1$ and $\delta_2$. **b** Models originating from the basic logic of mutualism yield diverse coexistence criteria. Each line represents the generation of a model from the basic mutualism logic and branching represents different implementation details and system complexities. The circles represent the models and diverse coexistence criteria derived from these models. This process aims to reflect the diversity of mutualistic systems in nature. **c** A simple rule emerges at an appropriate level of abstraction. The lines represent the abstraction process that establishes $B(\theta) > \delta$ as the common structure shared by diverse models in panel b. $B(\theta)$ represents effective benefit and is a complex function of model parameters $\theta$, which include $\beta$ and $\varepsilon$. $B$ increases with increasing $\beta$ and decreasing $\varepsilon$. The heatmap is generated using Eq. (4). $\delta$ is the stress experienced by one population. $r_m$ is growth rate measurement. Note that the color bar is dimensionless, and it is the same for all following color bars. **d** Intuitive interpretation of the simple rule. The effective benefit $B$ must overcome stress $\delta$ for the system to coexist. Solid black line represents coexistence boundary and dashed black line represents baseline fitness level with the absence of partner. Blue represents a $B$ that is greater than $\delta$ (coexistence) and yellow represents a $B$ that is smaller than $\delta$ (collapse). **e** $B/\delta$ can predict various system outcomes. If the two features $B$ and $\delta$ are known, many downstream predictions, both qualitative and quantitative, can be made. **f** Quantitative outcomes versus $B/\delta$. Simulation results show when $B/\delta > 1$, it is predictive of total density. Note that the points do not necessarily lie on a single curve, but a positive trend is well-maintained. Other quantitative outcomes also follow similar positive trends when plotted against $B/\delta$

**Table 1 Examples of benefit, cost, and stress in diverse mutualistic systems**

| Category | Partners | Benefit | Cost | Stress |
|---|---|---|---|---|
| Transportation mutualism | Plants | Increased fecundity[62] | Seed consumption and energy loss[62, 63] | Limited spatial range for reproduction |
| | Seed dispersers or pollinators | Access to nutrient-rich food | Energy loss or by-product mutualism | Starvation |
| Protection mutualism | Plants | Increased fitness due to reduced consumption from herbivores | Energy loss or by-product mutualism | Consumption by herbivores and competing plants |
| | Ants | Increased access to nutrients and shelter | Energy loss or by-product mutualism | Lack of suitable nesting sites[64] |
| Nutritional mutualism | Bacterial and archaeal auxotrophs | Increased nutrient availability in the environment | Energy loss or by-product mutualism | Nutrient-poor environments |
| Nutritional mutualism | Corals | Higher rate of calcification and conservation of nutrient[65] | Reduced cover, growth and fecundity[66] | Nutrient-poor marine environment |
| | Algae | Better habitat and increased availability of inorganic compound[67] | Energy loss, possible restricted growth by coral[67] | Nutrient-poor marine environment |

are universal features of mutualistic systems (Table 1). In addition to benefit and cost, which are conventionally considered as the driving forces of mutualistic outcomes[29–31], we included stress to capture the reduction of baseline fitness of individual populations from their maxima. Although stress is not always explicitly acknowledged in previous models, evidence indicates that it is a determining factor of mutualistic outcomes[32–34]. Incorporating stress can thus provide a more complete picture of mutualistic behavior (see Supplementary Note 2.1 for the detailed reasoning). Note that, in this study, we aim to capture ecological population dynamics only, and do not explicitly include evolutionary dynamics.

To reflect the diversity of natural mutualistic systems, we systematically generated a total of 52 ordinary differential equation models based on this basic logic of mutualism with various implementation details (see Methods and Supplementary Note 2.2 for model assumptions). These implementation details are designed to comprehensively cover the various common and plausible forms of kinetic models that have been adopted in previous studies (see Supplementary Note 2.3 for a summary). Specifically, the models all revolve around the logistic growth equation but differ in the locations of $\beta$, $\varepsilon$, and $\delta$ in the logistic growth equations, enabling constant, linear and saturating effects of $\varepsilon$, as well as saturating effects of $\beta$. Our models also include complexities such as competition, asymmetry, and turnover rate. We only increased the model complexity to an extent that closed-form steady state solutions are obtainable (see Supplementary Notes 2.3–2.5 for model construction rationales and details).

We derived coexistence criteria for all 52 models by requiring the coexistence steady state to be real and positive. This allows us to find the inequality that governs the transition between coexistence and collapse. For example, the following simple mutualism model has five fixed points (model 21 in Supplementary Table 1):

$$\frac{dX_1}{d\tau} = \frac{1}{\varepsilon}X_1(1-X_1) - \frac{\delta}{\beta X_2 + 1}X_1. \qquad (1)$$

$$\frac{dX_2}{d\tau} = \frac{1}{\varepsilon}X_2(1-X_2) - \frac{\delta}{\beta X_1 + 1}X_2. \qquad (2)$$

The fixed point that represents coexistence is:

$$(X_1^*, X_2^*) = \left( \frac{\beta - 1 + \sqrt{(\beta+1)^2 - 4\beta\varepsilon\delta}}{4\beta}, \frac{\beta - 1 + \sqrt{(\beta+1)^2 - 4\beta\varepsilon\delta}}{4\beta} \right). \qquad (3)$$

For this fixed point to be real and positive, the following inequality must hold (see Supplementary Note 3.1 for details):

$$\frac{(\beta+1)^2}{4\beta\varepsilon} \geq \delta \, (\beta \geq 1). \qquad (4)$$

Using this approach, our derived criteria exhibit diverse structures (Fig. 1b, Supplementary Tables 1 and 2, also see Supplementary Note 3 and Supplementary Software 1). The diversity of our criteria is consistent with the diversity of criteria that already exist in the literature[17–21]. This diversity highlights the need to have a general rule, since the appropriate model formulation for a specific system is often unknown a priori and its selection can also be nontrivial[35].

Despite the diversity, at an appropriate abstraction level, however, all criteria follow a simple general form (Fig. 1c):

$$B(\boldsymbol{\theta}) > \delta, \qquad (5)$$

where $\boldsymbol{\theta}$ denotes model parameters including $\beta$, $\varepsilon$, asymmetry, turnover rate, and other model complexities. $B(\boldsymbol{\theta})$ represents the effective benefit produced through mutualistic interaction. Quantitatively, $B(\boldsymbol{\theta})$ increases with increasing $\beta$ and decreasing $\varepsilon$ and its structure differs depending on the specific model. $\delta$ represents stress; it is determined as $1 - r_m$, where $r_m$ is the growth rate of the population in the absence of its mutualistic partner and normalized by its maximum growth rate. The interpretation of our criterion is intuitive: mutualistic partners can coexist if the effective benefit exceeds stress, and the system collapses when the inequality is violated (Fig. 1d). Note that although alternative forms of the criterion may exist, Eq. (5) is the most intuitive and parsimonious form.

When both asymmetry and competitive interactions are incorporated, the models can also exhibit transitions between coexistence and competitive exclusion besides the transition between coexistence and collapse. Although both transitions are

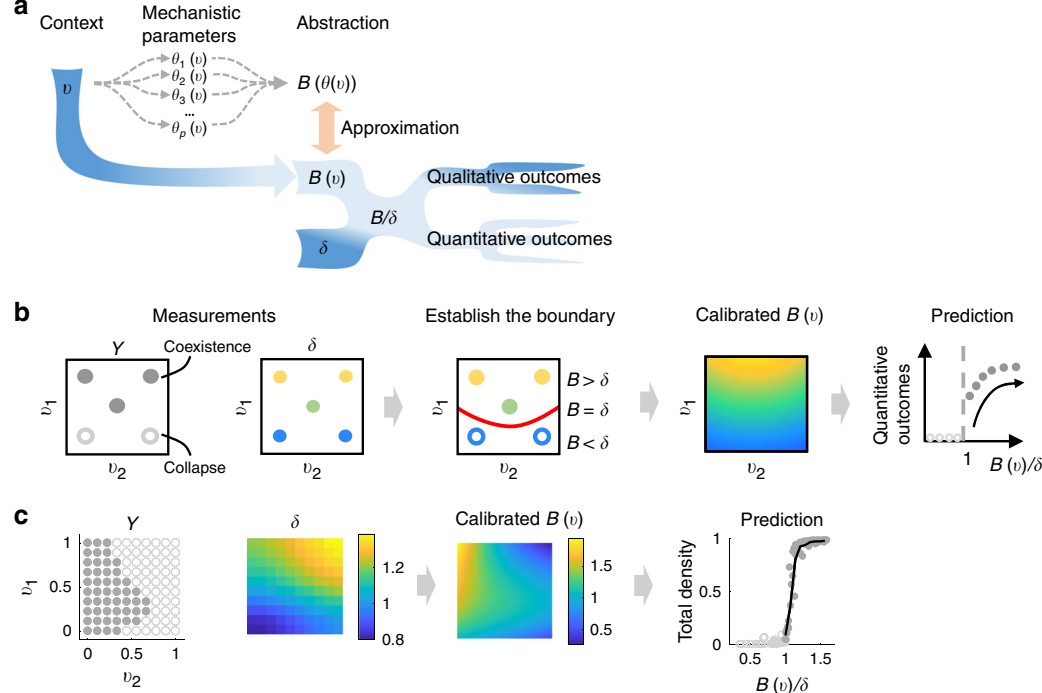

**Fig. 2** A streamlined approach to calibrate for an empirical $B(v)$. **a** The rationale behind the calibration procedure. Conventional approaches (denoted by dashed gray arrows) require quantifications of mechanistic parameters as functions of contextual variables $\theta(v)$ and finding the appropriate structure of $B(\theta)$ to construct $B(\theta(v))$. However, both steps are challenging and require case-by-case procedures. Instead, using qualitative outcomes of the system, we can calibrate for an empirical function $B(v)$ to approximate the true $B(\theta(v))$. $B(v)/\delta$ can then predict qualitative and quantitative outcomes. Dark blue indicates the data that are relatively easy to measure without requiring mechanistic understanding of the interaction. **b** A schematic demonstrating the mathematical basis of the calibration procedure. $v_1$ and $v_2$ represent two system variables. A circle represents an observation $i$ at a particular $v_i = (v_{i1}, v_{i2})$. Five observations are shown. $Y$ contains qualitative system outcomes for each observation. Closed circles indicate coexistence and open circles indicate collapse; the same notation scheme is used for all following figures. $\delta$ contains the measurement of stress for each observation (lighter colors indicate higher values). Using $v$, $Y$, and $\delta$, a boundary that separates the two outcomes can be established (the red curve). According to our simple rule, $B = \delta$ on the boundary; $B > \delta$ for coexistence and $B < \delta$ for collapse. Using these data and our simple rule, we can calibrate for a $B(v)$ which then enables the interpretation and prediction of system outcomes. Refer to Supplementary Movie 1 for a 3D visualization of the calibration. **c** Proof of principle using simulated data. Simulations were performed using a complex mutualism model that does not have an explicit form of $B(\theta)$ (see Supplementary Note 5.6). The input data set contains 100 observations. $\delta$ and calibrated $B(v)$ share the same axes with $Y$ (this applies to all following figures). $B(v)/\delta$ correctly classifies 97.2% of 2500 new data points. $B(v)/\delta$ is also predictive of total densities (only 100 data points are shown out of 2500). Black trace in the plot named "Prediction" represents binned averages of total density (this applies to all following figures). See Supplementary Note 5.7 for the detailed step-by-step calibration procedure

characterized by the loss of one or more populations, our model dynamic shows that collapse corresponds to lowered fitness of persisting partners, but competitive exclusion corresponds to an increased fitness of persisting populations. While many mutualistic systems also have competitive interactions[36–38], the transition to competitive exclusion cannot be generated by a mutualism interaction alone (see Supplementary Note 3.3.7 for more detailed discussion). Thus, we did not derive our criterion to predict competitive exclusion.

Beyond determining qualitative system outcomes (coexistence versus collapse), $B/\delta$ defines a general metric that is also positively related to quantitative mutualistic outcomes (Fig. 1e), such as final population density, probability of coexistence, and resistance to cheater exploitation. The predictive power of the metric is robustly maintained for both symmetric and asymmetric systems, including obligate and facultative mutualistic systems (Supplementary Figure 2a–f, Supplementary Note 4). Further, the theoretical prediction accuracy of our criterion is also robustly maintained in the presence of noise (Supplementary Figure 2g). The generality of the metric indicates that it is a general property of the class of mutualism models we have constructed and is a quantitative description of a core characteristic of mutualism. If

so, $B$ is a high-level feature that, along with $\delta$, provides a unifying framework for interpreting and predicting diverse mutualistic systems.

**A calibration approach to use the metric.** Quantification of both $B$ and $\delta$ are required to use the metric. Although $\delta$ is often easy to measure since it is a property of individual populations (see Supplementary Figure 3 for the general quantification procedure), quantification of $B$, which describes the interactions, is often challenging. Beyond the difficulties of selecting an appropriate structure for $B(\theta)$, quantification of its underlying parameters often requires nontrivial mechanistic characterizations, such as parameter fitting and specific biochemical assays. These mechanistic characterizations are especially challenging for cooperative traits, even in well-defined synthetic systems[39–41]. Applications of the criterion would thus be difficult for individual systems, let alone enabling streamlined applications for diverse mutualistic systems.

To bypass these challenges, we developed a calibration procedure to use qualitative outcomes to directly quantify $B$ as an empirical function of experimentally controllable variables ($v$), denoted by $B(v)$ (Fig. 2a). Specifically, $v$ consists of variables that

modulate system outcomes directly or indirectly, such as temperature, nutrient availability, genetic variation, initial seeding distance, and the extent of intermixing. $v$ measurements are often readily available, especially in laboratory settings where they are experimentally controlled independent variables. Thus, using simple measurements, we can approximate the true $B(\theta(v))$ that describes the diverse and complex interaction mechanisms without characterizing the specific mechanistic details. The calibrated $B(v)$ along with $\delta$, will serve as the basis for interpretation and prediction beyond initial data. Although the procedure requires initial measurements of *qualitative* outcomes, $B(v)/\delta$ can also provide predictive power for *quantitative* outcomes (Fig. 1e). Based on our theoretical analyses, we then expect $B(v)/\delta$ to be positively related to the final density, probability of coexistence, and cheater resistance. Further, $B(v)$ can be used to reveal how multiple system variables collectively alter the effectiveness of the interaction, which is a major challenge in studying context dependency of mutualistic outcomes[42].

We first defined the input–output relationship of the calibration procedure (Fig. 2b). Measurements of qualitative outcomes are denoted as $Y = [y_1, y_2, y_3, \ldots y_n]$ ($y_i = 1$ for coexistence and $-1$ for collapse; $i$ represents the index of an observation; $n$ represents the total number of observations). Measurements of $\delta$ for the same observations are denoted as $\delta = [\delta_1, \delta_2, \delta_3, \ldots \delta_n]$. Note that theoretically, quantification of $\delta$ for any partner is sufficient. However, choosing the partner with a larger dynamic range of $\delta$ is preferable since it can contain more information content. The context variables are denoted by $v = [v_1, v_2, v_3, \ldots v_n]$, where $v_i$ is a vector that contains the values of all system variables for observation $i$. With inputs $Y$, $\delta$ and $v$, we can establish a smooth boundary between coexistence and collapse described by $F(\delta, v) = 0$. To ensure $B > \delta$ for coexistence and $B < \delta$ for collapse, we constrain $F(\delta, v) > 0$ for coexistence and $F(\delta, v) < 0$ for collapse. Because $B = \delta$ is true at the boundary, we can deduce that $F(B, v) = 0$. According to the implicit function theorem, if $F(B, v) = 0$ is continuously differentiable, the output $B(v)$ is implied. A calibrated $B(v)$ can then enable downstream interpretation and prediction.

To implement the calibration, we used the support vector machine (SVM), a machine-learning algorithm for supervised classification (see Supplementary Notes 5.1–5.4 and Supplementary Software 1). Assuming continuity of $B(v)$, we used kernels that are separable in $\delta$ and $v$ to obtain $F(\delta, v) = 0$. We implemented linear, quadratic, cubic, and sigmoidal kernels to describe possible shapes of $B(v)$. Because there are infinite number of $B(v)$ that can provide equivalently high-classification accuracy, we ranked the $B(v)$ obtained from different kernels and different kernel parameters to find the $B(v)$ that are closer to the true $B(\theta(v))$ (Supplementary Figure 4a). The ranking method is established using simulated data where the true $B(\theta(v))$ is known, so that each $B(v)$ can be evaluated against $B(\theta(v))$ by coefficient of determination ($R^2$). We found that our procedure consistently optimizes for $R^2$ (Supplementary Figure 4b, c; see Supplementary Note 5.5 for figure details). The proper sample size for the calibration can be evaluated using the exponential decay of bias with increasing sample size[43] (Supplementary Figure 4d).

Using this procedure, we first tested whether $B(v)/\delta$ can be applied to mutualism models in which no explicit form exists for $B(\theta)$. To do so, we constructed an overwhelmingly complex two-population model with competition, partner-density-dependent cost, high Hill coefficient and asymmetric function structures (see Supplementary Note 5.6, Supplementary Figure 5a). Model parameters are functions of $v_1$ and $v_2$ (Supplementary Figure 5b). Using an input data set of 100 points (Supplementary Figure 5c), $B(v)/\delta$ correctly predicts coexistence versus collapse for 97.2% of

test data beyond the initial 100 data points. Detailed step-by-step calibration procedure is shown in Supplementary Note 5.7. As expected, $B(v)/\delta$ provides predictive power for quantitative outcomes including total population size (Fig. 2c), probability of coexistence (Supplementary Figure 5d) and resistance to cheater exploitation (Supplementary Figure 5e).

**Experimental applications in pairwise systems**. We next applied our framework to three experimental systems to test its applicability. As the first example, we engineered two synthetic mutualistic partners in Top10F' strain of *Escherichia coli*, denoted by $M_1$ and $M_2$ (Fig. 3a, Supplementary Figure 6a). In this system, stress is modulated by the concentration of Isopropyl β-D-1-thiogalactopyranoside (IPTG), which induces the expression of CcdB (a toxin). Independent from IPTG, anhydrotetracycline (aTc) induces quorum sensing (QS) modules in both strains to each produce a unique QS signal that triggers the production of CcdA (the antitoxin of CcdB) in the partner population. The production of aTc-induced expression of the QS module is responsible for the mutual benefit and can impose cooperation cost to both strains. Consistent with the circuit design, our experimental results demonstrated IPTG-mediated growth suppression and aTc-mediated mutual rescue (Supplementary Figure 6b).

We cocultured the two strains starting from the same initial density with different concentrations of IPTG and aTc, which are the two dimensions of $v$. The outcomes of coexistence and collapse are evident in the bimodal distribution of optical density (OD) at 32 h of culturing (Supplementary Figure 6c). $\delta$ can be quantified by treating monocultures with the same set of [IPTG] and [aTc]. We used $\delta$ for $M_2$ since it has a wider dynamic range (Supplementary Figure 6d). Using these data (Fig. 3b), we obtained a calibrated $B(v)$ (Fig. 3c). The confidence of $B(v)$ is evaluated by the consistency of the top five $B(v)$ and relative standard deviation of each $B(v)$ (Supplementary Figure 6e). Consistent with the circuit logic, $B(v)$ increases with increasing [aTc]. The calibration reveals that [IPTG] also modulates $B(v)$, which indicates unintended system complexities, such as QS cross-talk and unequal fitness of the two populations. We used cross-validation to evaluate how well new observations can be predicted. We found that $B(v)/\delta$ provides an average cross-validation accuracy of 96.8% for coexistence versus collapse and it is also predictive of total final density (Fig. 3d).

We then applied our procedure to data on a pair of *Saccharomyces cerevisiae* auxotrophs that is previously published[37]. In this system, one strain cannot produce tryptophan (Trp) and the other cannot produce leucine (Leu). The mutualistic interaction of this system is realized by the exchange of the two amino acids in cocultures (Fig. 3e). Because [Leu] is maintained as eight times of [Trp], we used [Trp] as one dimension of $v$ to represent overall concentration of supplemented amino acid. The authors also varied the ratio of initial densities which composes the other dimension of $v$ (Fig. 3f, Supplementary Figure 7a). All top five $B(v)$ reveal that intermediate ratios of initial density and increasing amino acid concentrations elevate $B(v)$ (Fig. 3g). However, at the highest level of supplemented amino acid ([Trp] = 16 nM, [Leu] = 128 nM), top-ranked $B(v)$ have qualitatively different trends, indicating a low confidence of $B(v)$ at high concentration (Supplementary Figure 7b). Although our criterion does not apply to the transition between coexistence and competitive exclusion, this high variability coincides with the system transitioning into competitive exclusion[37]. Nevertheless, $B(v)/\delta$ is still predictive of final densities with an average cross-validation accuracy of 95.0% (Fig. 3h). Furthermore, we explored using the concentration of

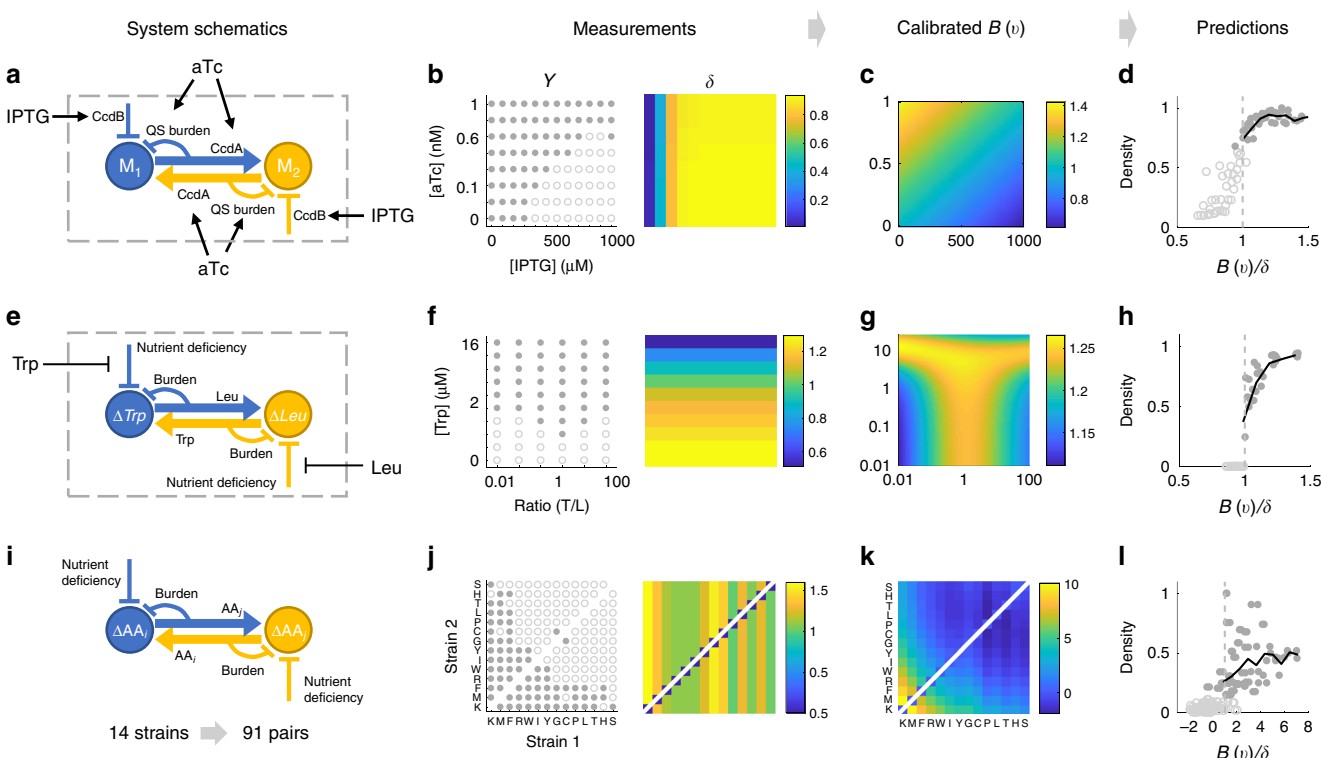

**Fig. 3** Application of the framework to experimental systems (see Supplementary Movies 2–4). **a** The QS-based mutualism system. IPTG modulates stress and aTc induces QS-mediated mutualistic interaction. **b** Measurements of coexistence and collapse and corresponding $\delta$ values. Coexistence and collapse are measured by coculturing the two strains starting from the same initial densities. $\delta$ is measured by OD of $M_2$ monoculture after 32 h of culturing. **c** Empirical calibration of $B(\boldsymbol{v})$. $B(\boldsymbol{v})$ reveals how [IPTG] and [aTc] together modulate the effectiveness of the interaction. **d** $B(\boldsymbol{v})/\delta$ is predictive of coexistence versus collapse and total final density. The x-axis range of [0.5–1.5] is used to highlight the transition (this also applies to other prediction plots). The trend continues to hold beyond this range. The y-axis represents normalized final cell density. **e** The pairwise yeast auxotroph system. The growth of both auxotrophs are suppressed in monocultures. With increasing Trp and Leu supplemented to the co-culture, the growth suppression can be alleviated. **f** The amount of supplemented amino acid and ratio of initial densities modulate system behavior. Only [Trp] is shown and [Leu] is eight times of [Trp]. The total initial density of the two strains are kept constant. Corresponding $\delta$ values are measured based on growth yield of $\Delta Leu$ monocultures, assuming $\delta$ is independent of initial density. **g** Optimal effective benefit occurs at an intermediate ratio of initial density. **h** $B(\boldsymbol{v})/\delta$ is predictive of normalized total cell number per culture well. **i** The 91 mutualism systems constructed by 14 engineered *E. coli* auxotrophs. Growth suppression is evident in their inability to survive individually in minimal medium. However, two auxotrophs can potentially survive through mutualistic interaction in a co-culture by exchanging amino acids. **j** System outcomes for all 91 pairs and $\delta$ for each of the 14 auxotrophs. Note that for one pair, the calibration is done twice with $\delta$ of either strain. **k** The calibrated $B(\boldsymbol{v})$ for *E. coli* auxotroph systems. **l** $B(\boldsymbol{v})/\delta$ is predictive of the normalized fold change of final total density relative to initial density

supplemented amino acid as a single system variable. $B(\boldsymbol{v})/\delta$ in this case can also predict the probability of coexistence as the ratio of initial densities varied (Supplementary Figure 7c).

In the third example, we applied our framework to previously published measurements of 14 engineered auxotrophic *E. coli* strains that compose 91 pairwise mutualistic systems[44] (Fig. 3i). The genetic context of the two partners varies while the growth environment was kept the same. The classification of coexistence versus collapse is based on the bimodal distribution of total density (Supplementary Figure 8a). $\delta$ of each auxotroph is determined based on final cell densities of monocultures when supplemented with different concentrations of its corresponding amino acid (Supplementary Figure 8b). We sorted the auxotrophs by the number of partners they coexist with to convert categorical indices into an ordinal scale. Thus, $\boldsymbol{v}$ is composed of ordinal rankings of the two strains and measurements of coexistence versus collapse and $\delta$ are both arranged accordingly (Fig. 3j). We used strain 1 as the reference strain for the calibration. The calibrated $B(\boldsymbol{v})$ generated a cross-validation accuracy of 91.8% and we verified that $B(\boldsymbol{v})/\delta$ is predictive of final total density

(Fig. 3k, Supplementary Figure 8c). We noticed a relatively high level of variability of total density when $B(\boldsymbol{v})/\delta > 1$, which can be due to system-specific properties that are not fully accounted for by mutualistic interactions.

**Applications in more complex settings**. In nature, mutualism can occur among three or more partners[45]. Thus, we tested our framework with simulations and experimental measurements of N-mutualist systems. Here, we show the calibration procedure with simulated data from a 5-mutualist system and found that the quality of the calibration results is well-maintained (Fig. 4a, Supplementary Figure 9a, Supplementary Note 6.1). The study that constructed the 14 auxotrophs[44] also presented all possible three-member double-auxotroph systems with the same set of amino acid deficiencies. Using the same procedure with a three-dimensional $\boldsymbol{v}$, where each dimension represents one amino acid the triplets are sharing, we found the predictor $B(\boldsymbol{v})/\delta$ provides an 89.3% cross-validation accuracy and remains predictive of the total density, which indicates the scalability of the framework in experimental settings (Fig. 4b, Supplementary Note 6.2).

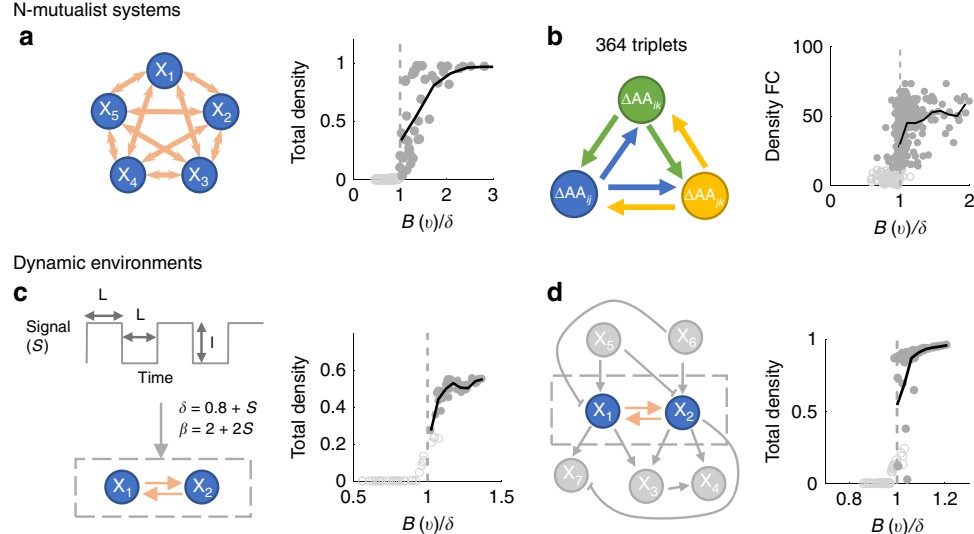

**Fig. 4** Application of the metric in complex settings. **a** A simulated mutualistic system with five partners. Parameters in the model are functions of two independent variables. Using 100 data points, we obtained a $B(\boldsymbol{v})$ through clibration. $B(\boldsymbol{v})/\delta$ successfully predicts coexistence versus collapse for 98.6% of a new set of 2500 data points (100 points are shown) and it is also predictive of total density. **b** Experimental auxotrophic triplets that are comprised of 91 *E. coli* double-auxotrophs with the same set of amino acid deficiencies in Fig. 3i–k. This experimental validation demonstrates the generality of our framework beyond pairwise interactions. **c** A system that is modulated by an oscillatory signal. The oscillatory signal $S$ is described by $I$ (intensity) and $L$ (duration). The signal modulates $\delta$ and $\beta$ temporally. $I$ and $L$ are the two system variables used in calibration. The procedure achieves a prediction accuracy of 97.3% for new data. **d** A simulated mutualistic system that coinhabits with 5 other populations. $X_1$ and $X_2$ are the mutualistic partners. $X_3$–$X_7$ are bystander populations that either modulate or are modulated by $X_1$ and $X_2$. $B(\boldsymbol{v})/\delta$ successfully predicts 92.3% of new data in this example

Additionally, we hypothesized that $B(\boldsymbol{v})$ calibrated for pairwise interactions can be used to directly construct a metric for three-member systems, since theoretical analysis shows that $n$-member $B(\boldsymbol{\theta})$ can be approximated by pairwise $B(\boldsymbol{\theta})$ (Supplementary Table 2). We assume $B$ of a three-member system is the average of $B$ for all three combinations of its underlying two-population systems and the same is true for $\delta$. The constructed $B/\delta$ for three-member systems can explain 80.8% of system outcomes (Supplementary Figure 9b). This result suggests the possibility of directly extending $B$ and $\delta$ from simple systems to more complex systems without further calibration.

Beside static environments, mutualistic systems can also inhabit dynamic environments where they experience fluctuating physical and chemical cues or cohabitate with other populations. We verified that the theoretical criterion generally holds in both cases (Supplementary Figure 10a). However, the transition between collapse and coexistence does not strictly occur at 1, which further advocates for the necessity of the calibration procedure. With simulated data, we carried out the calibration procedure and verified that the applicability of our framework is well-maintained (Fig. 4c, d, Supplementary Figure 10b, Supplementary Notes 6.3 and 6.4). The robustness of the framework suggests that it can be used to study microbial communities, of which advancements in both interpretation and prediction are in demand[46].

Mutualistic systems can generate complex temporal dynamics. For example, a mutualistic system that exhibits limit cycles has been previously reported[47]. The system is comprised of two *E. coli* strains that one is resistant to ampicillin and the other is resistant to chloramphenicol. When mixed together, the two strains deactivate the antibiotic they are resistant to and provide protection to the other sensitive strain (Supplementary Figure 11a). With periodic dilution, the relative abundance of the two strains oscillate over time. We used the model published in this previous study to simulate the growth dynamics at different antibiotic concentrations (Supplementary Figure 11b). Despite the oscillatory dynamics (Supplementary Figure 11c), our calibration procedure still reliably predicts coexistence versus collapse and provides an average cross-validation accuracy of 96.8% (Supplementary Figure 11d, e).

## Discussion

The immense complexity and diversity of biological systems is intriguing and inspires the exploration of mechanistic details. However, these details can distract us from simple rules that emerge at a higher level. By abstracting away from low-level details, many simple rules for biological systems have been developed to enhance our understanding and provide predictive power. A classic example is the Hamilton's rule, which states that a cooperative trait will persist if $\frac{c}{b} < r$, where $r$ is the relatedness of the recipient and the actor; $b$ is the benefit gained by the recipient; and $c$ is the cost to the actor. More recent examples include linear correlations underlying cell-size homeostasis in bacteria[48–50], ranking of quorum sensing modules according to their sensing potential[51,52], and the growth laws resulting from dynamic partitioning of intracellular resources[53,54].

Beyond establishing another simple rule, by focusing on mutualistic interactions, we also demonstrated that one can purposefully seek an appropriate abstraction level where a simple unifying rule emerges over system diversity. If this rule anchors in the basic definition of a type of system, it can then be applied to diverse systems of the same type. Beyond microbial systems that we tested, our criterion in principle can also be applied to other systems of larger or smaller scales that share the same logic.

In our demonstrations, we have focused on the analysis of homogenous systems. To account for the spatial dimension[55,56], one can incorporate spatial variables into our framework as context variables ($\boldsymbol{v}$). For example, the context variable can be the seeding distance of two partners or the degree of intermixing of the seedings. Calibrated $B(\boldsymbol{v})$ will then be dependent on these

spatial variables. Alternatively, the criterion can be applied to local segments where the homogeneity assumption is appropriate. In general, it remains an open question whether and to what extent our approach would be applicable if the mutualistic system becomes much more complex than what we have tested, such as systems consisting of multiple attractors that all correspond to coexistence.

Although simple general rules in biology can be powerful tools, their applicability to experimental systems can be limited by the difficulties in associating the abstracted parameters to lower-level mechanistic details and quantifying these details experimentally. This is evident in the application of Hamilton's rule to experimental systems[39–41]. For many inequality-based simple rules that have been proposed and established[10,57,58], our calibration procedure provides a generally applicable tool to apply these rules directly to experimental systems. If one side of the inequality and some final outcomes can be measured or have been observed historically, the other side can be calibrated as an empirical function. Although our procedure cannot further dissect the empirical function into specific mechanistic parameters, the function can serve as an overall summary of the underlying mechanistic details while bypassing the requirement of characterizing them individually. Our approach thus can enable the downstream interpretation and prediction by these simple rules with readily accessible measurements.

## Methods

**Model development**. We built mutualism models based on four key assumptions:

(a) Benefit shall increase growth rate or carrying capacity and is positively dependent on partner density.
(b) Cost shall decrease growth rate or carrying capacity.
(c) Stress shall produce negative growth of populations at some parameter combinations.
(d) Negative growth of a population shall be potentially counteracted by benefit provided by a partner, but further strengthened by cost.

See Supplementary Note 2 for detailed reasoning and implementation of each assumption.

**Criteria derivation**. We calculated the analytical solutions of fixed points of the 52 models using MATLAB R2017a. Then we identified the fixed points that represent stable coexistence. The coexistence criteria are derived by ensuring the fixed points are real positive numbers. We can then rearrange the inequality to have $\delta$ on one side. The other side of the inequality is then an expression of other parameters, which is expressed as $B(\theta)$. All criteria were verified using time course simulations. More details are presented in Supplementary Note 3. The MATLAB code of the models and the derivation and testing process is included in the Supplementary Software 1.

**Calibration procedure using SVM**. We used SVM algorithms in MATLAB to implement the calibration. The input data are formulated as following:

$$\text{Label of coexistence versus collapse}: \quad Y = [y_1, \cdots, y_i, \cdots, y_n]. \quad (6)$$

$$\text{System variables}: \quad \boldsymbol{v} = [\boldsymbol{v}_1, \cdots, \boldsymbol{v}_i, \cdots, \boldsymbol{v}_n]. \quad (7)$$

$$\text{Stress of the reference population}: \quad \boldsymbol{\delta} = [\delta_1, \cdots, \delta_i, \cdots, \delta_n]. \quad (8)$$

In Eqs. (6)–(8), $n$ represents total number of observations and each index represents one observation. $Y$ takes values of 1 or $-1$, which represent coexistence versus collapse for each observation. $\boldsymbol{v}$ contains the coordinates where observations are obtained and $\boldsymbol{v}_i$ is a vector of which each element represents a context variable. For a system with two system variables, $\boldsymbol{v}_i = (v_{i1}, v_{i2})$. $\boldsymbol{\delta}$ contains the stress level of the reference population for each observation $i$. $\boldsymbol{v}$ and $\boldsymbol{\delta}$ are first standardized to $\boldsymbol{v}^s$ and $\boldsymbol{\delta}^s$ that have mean of 0 and standard deviation of 1. For simplicity of presentation, the following $\boldsymbol{v}$ and $\boldsymbol{\delta}$ are standardized.

We designed kernels that have additive separability between $\boldsymbol{v}$ and $\boldsymbol{\delta}$, which can be expressed in a general form:

$$K\left\langle \left[\boldsymbol{v}_i, \delta_i\right], \left[\boldsymbol{v}_j, \delta_j\right] \right\rangle = K_v\left\langle \boldsymbol{v}_i, \boldsymbol{v}_j \right\rangle + k_\delta\left(\delta_i \cdot \delta_j\right). \quad (9)$$

$K_v$ is the kernel that dictates the shape of the empirical function of $B$ and $k_\delta$ is a kernel parameter. The predictor trained using SVM is:

$$f([\boldsymbol{v}, \delta]) = \sum_i \alpha_i y_i K_v\langle \boldsymbol{v}_i, \boldsymbol{v} \rangle + k_\delta \delta \sum_i \alpha_i y_i \delta_i + \lambda_0. \quad (10)$$

$\alpha_i$ is the weight of observation $i$, and $\lambda_0$ is the bias term. Both $\alpha_i$ and $\lambda_0$ are optimized by the SVM algorithm. $y_i$, $\boldsymbol{v}_i$, and $\delta_i$ are input values for observation $i$.

According to our criterion, we know that $B = \delta$ when $f([\boldsymbol{v}, \delta]) = 0$. We can then derive $B_0(\boldsymbol{v})$, a primitive function of $B$, from Eq. (10):

$$B_0(\boldsymbol{v}) = \delta = \frac{-\sum_i \alpha_i y_i K_v\langle \boldsymbol{v}_i, \boldsymbol{v} \rangle - \lambda_0}{k_\delta \sum_i \alpha_i y_i \delta_i}. \quad (11)$$

To obtain $B(\boldsymbol{v})$, $B_0(\boldsymbol{v})$ is then adjusted for directionality and rescaled back according to mean and standard deviation of the original $\boldsymbol{\delta}$ measurements.

To find the optimal $B(\boldsymbol{v})$, we used linear, quadratic, cubic, and sigmoidal kernels with a range of kernel parameters to train many different $B(\boldsymbol{v})$. The optimal $B(\boldsymbol{v})$ has the lowest overall cross-validation classification loss and bootstrapped variance. A final $B(\boldsymbol{v})$ is then used along with $\boldsymbol{\delta}$ measurements for interpretation and prediction. See Supplementary Note 5 for the detailed calibration method. For graphical representations of the step-by-step procedure see Supplementary Figure 4a and specifically Supplementary Note 5.7. We also have included in the Supplementary Software 1 the calibration procedure and sample data sets.

**QS-based mutualism strains**. The two strains were constructed based on circuit components from a synthetic predator-prey system[59,60]. Both populations carry two plasmids. Briefly, $M_1$ carries plasmids identical to the predator plasmids, denoted A1 for the module carrying *ccdA* (*tet* promoter[61] driving *luxR* and *lasI* followed by *lux* promoter driving *ccdA*) and B1 for the module carrying *ccdB* (*Lac* promoter[61] upstream of *ccdB* followed by *tet* promoter upstream of *gfp*). To construct $M_2$, A1 was used as backbone. To obtain orthogonal communication, KpnI and NotI restriction digest cloning was used to replace *luxR/lasI* genes from A1 with *lasR/luxI* genes from the previously published prey plasmid (consisting of pLac lasRluxI CcdB (Kan$^R$, p15A ori)). Reporter plasmid B1 is from[59]. To construct B2, enzymes XhoI and KpnI were used to replace the *tet* promoter on prey plasmid with the *ccdB* module from B1. All $M_1$ and $M_2$ plasmids were verified using restriction digest and sequencing.

**Growth conditions of QS-based synthetic system**. The experiments of QS-based mutualistic system were done in 96-well microtiter plates. PH-buffered M9 medium (M9 salt supplemented with 1 mM thiamine, 0.2% casamino acid, 0.4% glucose, 2 mM MgSO$_4$, 0.1 mM CaCl$_2$ and buffered with 100 mM MOPS with PH adjusted to 7.0) was used. Totally, 50 µg/ml kanamycin and 100 µg/ml chloramphenicol were added to the culture to maintain plasmids.

To measure circuit function, 4 ml LB media in a 14 ml culture tube was inoculated from single colony and incubated overnight at 37 °C at 250 r.p.m. The optical density is adjusted to 0.5 in M9 media (measured at 600 nm with TECAN microplate reader) before use. Cocultures are created by mixing both strains in a 1:1 volume ratio. The culture is then diluted $10^6$-fold and cultured in 200 µL batch culture at 30 °C in TECAN plate reader to record OD for 32 h with 10 min between each reading. The inducers were added to the media at the beginning with cell culture.

**Code availability**. The code used for data generation and/or analysis in the study are available as Supplementary Software 1.

## Data availability
The datasets generated during and/or analyzed during the study are available in the Supplementary Materials.

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

## Acknowledgments

We thank Tim Hoek and Jeff Gore for providing data used for analysis in Fig. 3f–h and Michael Mee and Harris Wang for sharing the data used for analysis in Fig. 3j, k. We also thank Yu Tanouchi, Lawrence David, Wenying Shou, Nan Luo, Yangxiaolu Cao, Carolyn Zhang, Ryan Tsoi, Teng Wang, and Shangying Wang for constructive inputs. This work is partially supported by grants from US National Institutes of Health (L.Y.: R01GM098642 and R01GM110494), National Science Foundation (L.Y.: MCB-1412459, C.L.: DEB 1257882, S.M.: DMS 17-13012, S.M.: ABI 16-61386, and S.M.: DMS 16-13261), Office of Naval Research (L.Y.: N00014-12-1-0631), Army Research Office (L.Y.: W911NF-14-1-0490), Human Frontier Science Program (S.M.: RGP0051), and a David and Lucile Packard Fellowship (L.Y.).

## Author contributions

F.W. conceived the research, designed and performed both modeling and experiments, interpreted the results, and wrote the manuscript. A.J.L. constructed the synthetic circuit, assisted with experimental design and manuscript revisions. D.N. assisted with modeling, results interpretation, and manuscript revisions. C.L. assisted with establishing the general relevance of the criterion and manuscript revisions. S.M. assisted with establishing the calibration process and manuscript revisions. L.Y. conceived the research, assisted in research design, data interpretation, and wrote the manuscript.

## Additional information

**Competing interests:** The authors declare no competing interests.

