## [Peer Review File · Nature Communications]

Reviewers' comments:

Reviewer #1 (Remarks to the Author):

By analyzing fixed points of several differential equation based models of mutualisms, Wu et al. have proposed a simple rule for coexistence of mutualistic species, based on generalized notions of benefit and cost associated with mutualism and the stress experienced by populations in the absence of mutualistic interactions. The rule states that coexistence of mutualistic partners requires the effective benefit to be larger than the stress. The authors have further devised a machine learning based calibration method to estimate the effective benefit based on measurements of a set of "context" variables. By applying this method to data obtained from simulations, previous experiments as well as a new experiment on synthetic mutualistic toxin-antitoxin producing strains of *E. coli* the authors show that the effective benefit B can be used to make quantitative predictions of population size in mutualistic systems, without prior knowledge of mechanistic details. From a conceptual standpoint, and given the class of models studied by the authors, I am not sure if I find the "rule" postulated here (namely $B(\theta) > \delta$) to be surprising. However, I find the fact that the effective benefit B can be extracted from a set of independent observables without any knowledge of the actual benefit and cost associated with mutualism to be interesting and useful. Given that mutualisms are indeed observed in a diverse array of ecosystems, I think there is sufficient merit in devising predictive schemes such as the one developed in this paper. I therefore feel that the paper could be suitable for publication in *Nature Communications*.

Major comments:

- 1) My primary concern with the paper is how difficult I found it to understand the procedure from the main text, despite the fact that I am very much part of the target audience. Might it be possible to use a very simple model when describing the initial motivation (stress, benefit, etc) and then to later show that the approach can be generalized to the wide class of models?
- 2) Is it fair to say that the fact that the authors' approach works means that all of these mutualisms are effectively one-dimensional? Does this mean that there is a separation of timescales, or is that not necessary? Some more discussion of when the approach will fail would perhaps clarify the limits as well as why the procedure works at all.
- 3) My primary concern with the procedure is that it entirely based on the dichotomy of coexistence and collapse, and appears to rely on either of these two states being globally stable fixed points of the system for a given set of parameters. As the authors themselves have pointed out, in their analysis of the yeast cross-feeding mutualism data, their method fails in the limit of high amino acid concentrations, where the global fixed point corresponds to competitive exclusion. A similar problem might also occur if multiple stable attractors coexist, because in that case, the outcome of the mutualism would depend strongly on the initial conditions.
- 4) It is not clear to me whether the method works if the attractor is a limit cycle instead of a fixed point, as is the case with the bacterial cross-protection mutualism (Yurtsev, Conwill and Gore, PNAS, 2016). It may still be the case that the rule makes qualitatively correct predictions as to whether the mutualism survives or not, but I would expect the quantitative predictions to be more erratic. It would be helpful if the authors could elaborate on this.

Minor comments:

- 1) It would be nice to include a small discussion on how the calibration method could potentially be extended to include spatial structure. Is it possible to incorporate space as an additional variable in the set of "context" variables, or would it require the supervised learning algorithm to be run on a class of models that explicitly include spatial structure?

Reviewer #2 (Remarks to the Author):

While my primary expertise is in the evolutionary analysis (and modelling) of mutualism, I very much enjoyed reading this ambitious paper on ecological dynamics of mutualistic interactions. It has a big aim: to find general rules predicting population dynamics across (almost) all mutualistic systems, particularly coexistence and collapse conditions, as well as quantitative predictions (density, coexistence probability, time to cheating take-over).

In ecology and evolution, we can sometimes be too affected by what Queller called the 'tyranny of detail' (Am Nat. 2017 Apr; 189(4): 345-353), to even attempt finding such general rules. Abstract, higher-level (coarse-grained, as the authors call them) rules, can be very useful in guiding research, however, and I really like how the authors try to nevertheless find them.

To do so, they analyse a range of 81 different mutualisms models, based on a large set of ODEs, and aiming to reflect (most of the) diversity of mutualistic systems out there (more later on this). Deriving coexistence criteria these models, they come up with a general rule which predicts mutualism coexistence as a function of the effective benefit and stress experienced by the mutualistic partners (Eq 1). They then proceed to show how these parameters, or approximations of them, could be empirically measured and do so using a new experiment, simulations and a number of previously published (microbial) datasets.

The general rule the authors derive seems novel to me but makes a lot intuitive sense, and the application to simulations and particularly real-life datasets are both informative and convincing. Together they make for a convincing analysis, and a very useful contribution, with many potential applications in the study of mutualisms. The paper is also well written and overall a pleasure to read, and the code to replicate all theoretical analyses and simulations has all been made available.

However, I have one major concern, which I feel the authors need to address because it has the potential to substantially affect the claimed generality of these results.

As I understand it, the various models analysed by the authors, all have in common that benefit is 'positively dependent on partner density' (Third line methods section, Section IIB Supplementary), although this benefit is bounded (section D Sup info, near equation II.8). However, while there are certainly situations in which this is undoubtedly true (for instance, in many of the lab microbial interactions studied empirically in this work), I am not necessarily convinced that such positive dependence on partner density hold generally among mutualistic interactions. After all, there is lots of evidence from natural systems of saturating benefits from increasing partner densities, and indeed also of shifts to negative effects beyond certain densities (e.g. Anderson and Midgley, 2007; Morris, Vázquez and Chacoff, 2010; Vannette and Hunter, 2011; Geib and Galen, 2012; Palmer and Brody, 2013 -> refs at the bottom of this review; the DeAngelis, Holland & Bronstein 2002 paper cited by the authors also gives a few examples). Such saturation of benefits makes a lot of intuitive sense: at some points all the flowers in a plant population are pollinated, or all the herbivores chased away by protective ants. Additional ants are then at best neutral, and potentially a fitness cost (see Palmer 2013).

The authors do mention previous work on saturating benefits in their supplementary information (in the section where they discuss previous models and again when they discuss their own modelling approach), for instance discussing some of the Holland & DeAngelis work on this, but as I understand it they don't incorporate any potential saturating effects (or even shifts to negative effects a high-densities) in their models. Of course, with some exceptions, we don't generally know to what extent real-life mutualisms typically experience (bounded) positive benefits from increased partner-densities as in the current work versus when saturating or even negative density effects start to appear. However, given that we have good evidence (and theoretical reasons) to think that they may commonly exist, this could undermine the claim of generality in this paper. Consequentially, I would like to see the authors either (i) include such effects in their models and

analyse their impact, or (ii) convincingly explain and show why this concern is not relevant and their conclusions would hold even under saturating benefits, or (iii) scale back the claim of generality and clearly indicate that their results may not (all) apply to mutualisms with saturating effects, and are only directly applicable to the (much?) smaller subset of mutualisms with strict positive density effects.

I also have some more smaller remarks:

- Please include line numbers throughout for a potential resubmission, including for the supplementary. Referencing sections of the manuscript is very cumbersome without them.
- I think it would be important to highlight more explicitly in the main text that the models analysed here are all ecological models, and do not include potential evolutionary responses. It's fine to not include evolutionary dynamics, but it's important for the reader to be aware of this limitation, particularly given that eco-evolutionary dynamics could actually be important, particularly in many of the microbial mutualistic systems considered here.
- Page 8 - Experimental Application of the metrics: for the first experimental application of the metric, I didn't fully understand what, if anything, the model equivalent of aTc is. The stress is imposed/modulated by IPTG, if I understand correctly, so is this just some external trigger of Quorum-sensing, independent of stress, that for whatever reason is required in this system? Would we not want QS to be triggered by the stress/stress-inducer itself? Some clarification for the reader would be helpful here.
- Main text table: I don't very much like cancer as an example of mutualism here. I know that the authors are using a slightly wider definition of mutualism here of two populations providing reciprocal benefits (first sentence of the ms), but most typically the term is used for situations where these populations are also of different species. I would suggest instead using an interspecific protection mutualism (e.g. ant-plant).
- SI page 3 "In addition, although population collapse [...]the fitness of the benefit-receiver decreases with increasing partner density, which is contradictory to the basic logic of mutualism." -
> I appreciate that given the definition of 'mutualism', at the stage where effects are negative the interaction is no longer strictly speaking a mutualism, but following my above general remark I wouldn't describe this effect as being contradictory to the basic logic of mutualism. The interaction at some density having a negative effect and then no longer being a mutualism is no way contradictory to it being a mutualist in other conditions, which I think what the models discussed here were trying to analyse.

Best wishes,
Dr. Gijsbert Werner
University of Oxford
[\Signed]

References cited in this review:

- Anderson, B. and Midgley, J. J. (2007) 'Density-dependent outcomes in a digestive mutualism between carnivorous *Roridula* plants and their associated hemipterans', *Oecologia*, 152(1), pp. 115–120. doi: 10.1007/s00442-006-0640-8.
- Geib, J. C. and Galen, C. (2012) 'Tracing impacts of partner abundance in facultative pollination mutualisms: from individuals to populations', *Ecology*, 93(7), pp. 1581–1592. doi: 10.1890/11-1271.1.
- Morris, W. F., Vázquez, D. P. and Chacoff, N. P. (2010) 'Benefit and cost curves for typical pollination mutualisms', *Ecology*, 91(5), pp. 1276–1285. doi: 10.1890/08-2278.1.
- Palmer, T. M. and Brody, A. K. (2013) 'Enough is enough: the effects of symbiotic ant abundance on herbivory, growth, and reproduction in an African acacia', *Ecology*, 94(3), pp. 683–691. doi: 10.1890/12-1413.1.
- Vannette, R. L. and Hunter, M. D. (2011) 'Plant defence theory re-examined: nonlinear expectations based on the costs and benefits of resource mutualisms', *Journal of Ecology*, 99(1), pp. 66–76. doi: 10.1111/j.1365-2745.2010.01755.x.

Response to reviewers' comments

Reviewer #1 (Remarks to the Author):

By analyzing fixed points of several differential equation based models of mutualisms, Wu et al. have proposed a simple rule for coexistence of mutualistic species, based on generalized notions of benefit and cost associated with mutualism and the stress experienced by populations in the absence of mutualistic interactions. The rule states that coexistence of mutualistic partners requires the effective benefit to be larger than the stress. The authors have further devised a machine learning based calibration method to estimate the effective benefit based on measurements of a set of "context" variables. By applying this method to data obtained from simulations, previous experiments as well as a new experiment on synthetic mutualistic toxin-antitoxin producing strains of *E. coli* the authors show that the effective benefit B can be used to make quantitative predictions of population size in mutualistic systems, without prior knowledge of mechanistic details. From a conceptual standpoint, and given the class of models studied by the authors, I am not sure if I find the "rule" postulated here (namely $B(\theta) > \delta$) to be surprising. However, I find the fact that the effective benefit B can be extracted from a set of independent observables without any knowledge of the actual benefit and cost associated with mutualism to be interesting and useful. Given that mutualisms are indeed observed in a diverse array of ecosystems, I think there is sufficient merit in devising predictive schemes such as the one developed in this paper. I therefore feel that the paper could be suitable for publication in *Nature Communications*.

We thank the reviewer for recognizing the relevance and significance of our work and for constructive suggestions and comments.

Major comments:

1) My primary concern with the paper is how difficult I found it to understand the procedure from the main text, despite the fact that I am very much part of the target audience. Might it be possible to use a very simple model when describing the initial motivation (stress, benefit, etc) and then to later show that the approach can be generalized to the wide class of models?

We thank the reviewer for this constructive comment. We have included in the main text a simple mutualism model to demonstrate the derivation process (page 5 line 78-84). In addition, to further clarify the calibration procedure, we have included a new Extended Data Figure 5, which provides a step-by-step illustration of the calibration procedure using the simulated data of a complex mutualism model. This figure is shown here as Fig. R1.

Fig. R1: Detailed simulation and calibration procedure using a complex model.

- a. The model equations of a complex mutualistic system, which cannot be easily solved analytically.
- b. Model parameters as functions of system variables v_1 and v_2 . For an experimental system, v_1 and v_2 would correspond to experimentally controllable parameters, which could affect multiple mechanistic parameters simultaneously.
- c. Simulated time courses with different v_1 and v_2 values. The group on the left simulates cocultures (blue: X_1 and red: X_2). The group on the right simulates monoculture, where $X_2 = 0$. The growth rates of the monoculture can then be used to calculate δ . We directly used the parameter value of δ in our procedure.
- d. Detailed procedure of the calibration process:

1. Data collection and preprocessing: Collect the measurements of coexistence vs. collapse (\mathbf{Y}) and stress (δ) on the domain of \mathbf{v} . Above a certain threshold of total final density, assign \mathbf{Y} a value of 1 and assign -1 if total final density is below this threshold. Standardize the values of δ and \mathbf{v} values to have mean of 0 and standard deviation of 1 and name the normalized vectors δ^s and \mathbf{v}^s . For simplicity of presentation, the following \mathbf{v} and δ are standardized. In this example, we have 100 observations ($n = 100$, 10 by 10) and two dimensions of \mathbf{v} ($\mathbf{v}_i = (v_{i1}, v_{i2})$).
2. Pick the top SVM models: Using \mathbf{Y} , δ and \mathbf{v} as inputs, generate 1600 different SVM models, each containing one of the four kernel functions (linear, quadratic, cubic, and sigmoidal, see SI section V.A) and one of 400 kernel parameter sets. For each kernel parameter, select values within a predefined range. The exact number of sets of kernel parameters used is not critical and can be adjusted depending on available computational resources.
 - 1) Calculate CV_{loss} : For each SVM model, it can then predict coexistence or collapse for each combination of δ and \mathbf{v} values. These predictions are used to calculate the 10-fold cross-validation loss (CV_{loss}).
 - 2) Calculate Var : To calculate the variance for each SVM model, sample the 100 observations 100 times with replacement and calculate a single bootstrapped instance of $B(\mathbf{v})$. Repeat this process 500 times to obtain a distribution of bootstrapped $B(\mathbf{v})$. Using this distribution, we can calculate the mean variance (Var) of each of the 1600 SVM models.
 - 3) Calculate $0.2Var + 0.8CV_{loss}$: We empirically examined the top five SVM models, which were select by the lowest $0.2Var + 0.8CV_{loss}$ values. The weights of Var and CV_{loss} were determined empirically using simulated data (see Extended Data Fig. 4c).
3. Train the top SVM models: Train each of the five SVM models with normalized input data to obtain the function that describes the boundary between the two classes. The functions can be expressed as:

$$f([\mathbf{v}, \delta]) = \sum_i \alpha_i y_i K_v(\mathbf{v}_i, \mathbf{v}) + k_\delta \delta \sum_i \alpha_i y_i \delta_i + \lambda_0$$

α_i is the weight of observation i , and λ_0 is the bias term for the SVM model. Both α_i and λ_0 are optimized by the SVM algorithm. K_v is the kernel function; k_δ is a kernel parameter; y_i , \mathbf{v}_i , and δ_i are input values for observation i . \mathbf{v} and δ are independent variables of the function. A positive value of $f([\mathbf{v}, \delta])$ for a new set of δ and \mathbf{v} predicts coexistence and a negative value predicts collapse.

4. Quantify the top $B(\mathbf{v})$ and assess its reliability: Quantify $B(\mathbf{v})$ from each SVM model by imposing $B(\mathbf{v}) = \delta$ at $f([\mathbf{v}, \delta]) = 0$. If $B(\mathbf{v})$ has the correct directionality, get the function of $B(\mathbf{v})$ by calculating

$$B(\mathbf{v}) = \left(\frac{-\sum_i \alpha_i y_i K_v(\mathbf{v}_i, \mathbf{v}) - \lambda_0}{k_\delta \sum_i \alpha_i y_i \delta_i} \right) \cdot var + mean$$

var and $mean$ are calculated using the original δ measurements. See SI section V.C. for how to adjust for the directionality of $B(\mathbf{v})$.

- 1) Quantify relative standard deviations (RSD) of top $B(\mathbf{v})$: The sampling process is the same as step 2.2). Iterate 10 times to get 10 bootstrapped $B(\mathbf{v})$ to quantify an RSD value on each (v_1, v_2) pair and construct a variability landscape of $B(\mathbf{v})$. The title of each $B(\mathbf{v})$ is the cross-validation accuracy of the model.
- 2) Quantify pair-wise consistencies of the top 5 models: To get R^2 of any two $B(\mathbf{v})$, the two scales of $B(\mathbf{v})$ are unified by first using linear fitting to adjust one $B(\mathbf{v})$ to conform to the scale of the other $B(\mathbf{v})$. R^2 is then calculated using the two $B(\mathbf{v})$ that have adjusted scales.
- 3) Pick the best $B(\mathbf{v})$: The first $B(\mathbf{v})$ is picked in this case since all top 5 have similar RSD and are all highly consistent.
5. Use $B(\mathbf{v})$ in downstream predictions: Use the best $B(\mathbf{v})$ to construct the metric $B(\mathbf{v})/\delta$. $B(\mathbf{v})/\delta$ is indeed positively related to final total density in this example.

The same procedure is applied to the analysis of other simulated data and experimental data. The different sets of data only differ in terms of how they are generated and what v_1 and v_2 correspond to.

- e. The calibrated $B(\mathbf{v})$ along with δ is predictive of probability of coexistence. To get the probability of coexistence at each (v_1, v_2) , we ran 100 simulations with 100 different ratios of initial densities while keeping the total initial density the same.
- f. $B(\mathbf{v})/\delta$ is also predictive of how well the system can resist cheater exploitation. The y axis indicates the time the system can persist before the cheater populations compete out the cooperators.

2) Is it fair to say that the fact that the authors' approach works means that all of these mutualisms are effectively one-dimensional? Does this mean that there is a separation of timescales, or is that not necessary? Some more discussion of when the approach will fail would perhaps clarify the limits as well as why the procedure works at all.

We thank the reviewer for these insightful questions and observations. Our brief response is that yes, our claim is that, at a level of abstraction that is extremely general but still meaningful, it is possible to interpret and predict mutualism in a simple way with B and δ .

Response to "one-dimensional": Although by formulation, the underlying models are at least two dimensions (as summarized in SI Table 1 and 2), one can indeed interpret our criterion as a "one-dimensional" simplification and generalization of mutualistic systems. The emergence of our criterion indicates that, along a properly chosen trajectory (dictated by the shape of estimated B), certain coarse-grain outcomes of a mutualistic system (coexistence versus collapse, total density, probability of coexistence and persistence to cheater exploitation) approximately align with each other and with B/δ in a monotonic manner.

Response to "separation of time scale": From our derivation, the separation of time scales does not appear to be critical. The mechanism-based criteria derived from specific models (SI Table 1 and 2) emerge from the steady-state solutions of these models. As such, the potential separation of time scales in these models is masked in the corresponding criteria.

Response to "when the approach will fail":

As demonstrated in the paper, our criterion and procedure work for the coarse-grained interpretation and prediction of the transition between mutualistic coexistence and collapse, total productivity, probability of coexistence, and time it takes for cheater exploitation. The criterion emerges from the analysis of steady-state solutions of two-population models. However, as indicated in the analysis in Figure 4, the criterion and procedure also work for higher-dimensional systems and for systems exposed to certain dynamic perturbations. These results further demonstrate that the steady-state assumption is not a critical assumption.

As noted below, our approach also works if a mutualistic system exhibits limit-cycle oscillations.

In general, it remains an open question whether and to what extent this approach would be applicable beyond the scope of mutualism and if the mutualistic system becomes much more complicated than what we have tested. We have included a sentence on page 13 (line 295-297) of the main text to discuss this limitation.

Response to “why the procedure works at all”:

The calibration procedure (based on SVM) we have developed works when we know *a priori* that a system is dictated by an inequality: in a general term, $X > Y$ dictates qualitative outcomes and X/Y is positively correlated with quantitative outcomes.

The theoretical foundation of our procedure is the class of mutualism models we have constructed and analyzed. The class of models share two key properties:

- a) a general criterion $B > \delta$ that underlies coexistence of two populations;
- b) B/δ is positively correlated with total density, probability of coexistence and resistance to cheater. These positive correlations between B/δ and quantitative outcomes are predicted by our class of models.

The reason why this class of models behave similarly, we think, resides in the assumptions of our models. Our assumptions (see Method section and SI section II.B) strive to find **the minimal description of mutualism**: two populations help each other to reduce the stress that each receives and impose a cost to itself. Stripping away the specificities of criteria derived from each individual model, our criterion can be considered as a general quantitative description of the assumptions that are shared by all models.

We have included our clarification and discussion of the limitations of our procedure in main text (page 6 line 116-118 and page 7 line 141-142).

3) My primary concern with the procedure is that it entirely based on the dichotomy of coexistence and collapse, and appears to rely on either of these two states being globally stable fixed points of the system for a given set of parameters. As the authors themselves have pointed out, in their analysis of the yeast cross-feeding mutualism data, their method fails in the limit of high amino acid concentrations, where the global fixed point corresponds to competitive exclusion. A similar problem might also occur if multiple stable attractors coexist, because in that case, the outcome of the mutualism would depend strongly on the initial conditions.

We thank the reviewer for these thoughtful comments.

Response to “global fixed points”: We regret the lack of clarity on this point in our original manuscript. Our procedure does not require the collapse and coexistence states to be global fixed points. For example, a simple mutualism model can have both the collapse state (0,0) and the coexistence state with the same parameter set. In our SI tables where we calculated the criterion for coexistence, we analyzed the conditions where the fixed points corresponding to coexistence are real and positive. This means that if the initial density is sufficiently high (above the separatrix), the two populations coexist. The populations collapse when the coexistence stable steady state does not exist.

Using the following model (eq. III.22-23 in Extended Data), we can generate vector fields that have multiple fixed points (Fig. R2):

$$\frac{dX_1}{dt} = \frac{1}{\varepsilon_1} X_1(1 - X_1 - X_2) - \frac{\delta_1}{\beta_2 X_2 + 1} X_1$$

$$\frac{dX_2}{dt} = \frac{1}{\varepsilon_2} \rho X_2(1 - X_1 - X_2) - \frac{\delta_2}{\beta_1 X_1 + 1} X_2$$

Fig. R2. Vector fields of different parameter sets using the model presented in SI eq. III.22-23. The two vector fields in red circles both have two stable steady states. The criterion for the symmetric case is presented in Extended Data Table 1 model #48; the criterion incorporating initial density is in Extended Data Table 2 model #83; the asymmetric case corresponds to Extended Data Table 2 model #84. The derivations of these criteria can be found in SI section III.

In addition, we have demonstrated that initial densities can be included in the criterion as a part of θ , beyond benefit (β) and cost (ε). An example criterion that includes initial densities is shown in Extended Data Table 2, criterion #83 (derivation of the criterion can be found in SI. Section III.C.2). Thus, if initial density is a variable, it can be incorporated into the θ term of our criterion $B(\theta)/\delta$. If initial density is a random variable, we have also found that the probability of coexistence can be predicted by $B(\theta)/\delta$ (Extended Data Figure 2a, e and Extended Data Figure 7c).

Response to “competitive exclusion”:

We apologize that we have not made it clear in our main text that our criterion does not describe the transition between coexistence and competitive exclusion. The winner in competitive exclusion increases in fitness when the loser is excluded. In contrast, collapse corresponds to the transition where mutualistic partners experience a reduction in fitness when a population is extinct (main text page 3, line 34). In simulations, we found that when there is no competition, there is no competitive exclusion. Thus, the transition from coexistence to competitive exclusion is a property of competition, and not of mutualism.

The various criteria we presented in SI Table 1-2 were not derived in a way to predict the transition between coexistence and competitive exclusion. However, by imposing the fixed point $(0, X_1^*)$ or $(X_2^*, 0)$ to be stable, future studies can use a similar approach to derive a general criterion that describes the transition between coexistence and competitive conclusion. Suppose the criterion is $C <$

δ , where C is a lumped term describing effective competition, we can still use our calibration procedure to calibrate for an empirical C .

The following model demonstrates the relationships between these two transitions (SI eq. III.30-31), which incorporates both asymmetry and competition. If we do not include competition in the model ($a = 0$), competitive exclusion is not present:

$$\begin{aligned}\frac{dX_1}{d\tau} &= \frac{1}{\varepsilon} X_1(1 - X_1 - aX_2) - \frac{\delta}{\beta X_2 + 1} X_1 - \delta_0 X_1, \\ \frac{dX_2}{d\tau} &= \frac{1}{\varepsilon} \rho X_2(1 - aX_1 - X_2) - \frac{\delta}{\beta X_1 + 1} X_2 - \delta_0 X_2.\end{aligned}$$

Using $\varepsilon = 1.2$, $\beta = 10$ and $\delta_0 = 0.1$, we generated the following phase diagram of system behaviors (Fig. R3).

Fig. R3. Both upper bound (blue) and lower bound (black) exist for δ when there is competition. The region below the black dots represents competitive exclusion. The region above the blue dots represents collapse. This diagram shows that transition into competitive exclusion and mutualistic collapse are two distinct boundaries. When there is no competition, the transition between coexistence and competitive exclusion is not present.

Since our criterion does not apply to the transition from coexistence to competitive exclusion, our prediction does not apply to the case when δ is lowered to an extent that competition starts to dominate mutualistic interaction. This applies to the yeast auxotroph example where δ becomes low when the two amino acids are supplied at high doses, as also noted by the reviewer.

In our initial submission, we included section III.C.7 in SI to discuss the distinctions between these two types of transitions. We have updated this section to include Fig. R3 to show the difference between the two types of boundaries and how the lower boundary of δ only occurs with competition. In addition, we have clarified this point in multiple places of our main text, including page 3 line 34, page 10 line 217-218 and added a new paragraph on page 6 line 102-109 to clarify this point.

Response to “multiple attractors”:

Indeed, it is unclear if our criterion is directly applicable when a system consists of multiple attractors, all corresponding to coexistence. This will likely depend on specific systems. For instance, for such a complex mutualistic system, it may still be possible to have a single boundary separating collapse and coexistence. In such an example, however, the quantitative predictive power of the estimated function will likely be limited. Further studies are needed to formulate such a model (in a relevant manner) and to test the applicability of our criterion and approach.

We included a discussion on page 13 line 295-297 of our manuscript to clarify this raised point.

4) It is not clear to me whether the method works if the attractor is a limit cycle instead of a fixed point, as is the case with the bacterial cross-protection mutualism (Yurtsev, Conwill and Gore, PNAS, 2016). It may still be the case that the rule makes qualitatively correct predictions as to whether the mutualism survives or not, but I would expect the quantitative predictions to be more erratic. It would be helpful if the authors could elaborate on this.

We thank the reviewer’s constructive comments and pointing us to this highly relevant paper. We have recaptured the simulation data presented in (Yurtsev, Conwill and Gore, PNAS, 2016), where a limit cycle exists for a pair of mutualistic bacterial strains. We have found that our calibration procedure still applies and provides high prediction accuracy.

By examining the paper suggested by reviewer, we found that the density of the two populations oscillate with a phase difference, where one increases, the other decreases. This complementary growth dynamic creates a rather stable total density overtime. Using the final timepoint of the final cycle for each condition, we classified the simulations into coexistence and collapse (Figure R4d). Using data in Figure R4, we conducted the calibration procedure and our procedure provides an estimation of effective benefit $B(\mathbf{v})$ and an average cross-validation accuracy of 96.8%. In this case, the overall densities do not have a wide distribution, so the quantitative predictive power of $B(\mathbf{v})/\delta$ were not tested (Figure R4e).

This example further illustrates the applicability of our criterion and calibration procedure to diverse mutualistic systems. We have included Figure R4 as Extended Data Figure 11 in and included a description of the system on page 12 (line 263-271) in the main text.

Fig. R4: Application of our approach to a mutualistic system displaying oscillatory dynamics.

a. System schematic and simulation procedure. N_1 is resistant to ampicillin and chloramphenicol imposes stress on it. N_2 is resistant to chloramphenicol and ampicillin imposes stress. Both strains can degrade antibiotics in the environment that they are resistant to. The simulations are done by looping through 12 cycles where the initial density of N_1 and N_2 are set to be 1/100 of the final densities of the previous cycle.

b. Model equations and parameter values for Fig. S7 in the original publication (Yurtsev, Conwill and Gore, PNAS, 2016).

c. Replication of Fig. S7 in (Yurtsev, Conwill and Gore, PNAS, 2016). The model used in this publication (eq. [3] in SI) also incorporates molecular details of the interaction mechanisms. Red traces represent the ampicillin resistant population and blue traces represent the chloramphenicol resistant population. The black dash lines represent total cell density.

d. Input data for our calibration procedure. δ is estimated according to Fig. 1D in (Yurtsev, Conwill and Gore, PNAS, 2016).

e. Calibration results show that our calibration procedure still provides high prediction accuracy. The total densities corresponding to coexistence do not have a wide distribution, so the quantitative predictive power of $B(\mathbf{v})/\delta$ were not tested

Minor comments:

1) It would be nice to include a small discussion on how the calibration method could potentially be extended to include spatial structure. Is it possible to incorporate space as an additional variable in the set of "context" variables, or would it require the supervised learning algorithm to be run on a class of models that explicitly include spatial structure?

We thank the reviewer for this insightful suggestion. We have not attempted to incorporate spatial structure into our criterion. Indeed, as the reviewer mentioned, a possible way to do so is to use different metrics of spatial structure as one dimension in the context variable. For example, the context variable can be the distance between the seedings of two partners or the degree of intermixing of the initial seedings. The calibrated $B(\mathbf{v})$ will then also be a function of these spatial contexts. We would expect decreasing seeding distance or increasing the extent of intermixing increases the benefit level and thus increases $B(\mathbf{v})$.

We also agree with the reviewer that an alternative is to explicitly include spatial structure in the models. When doing so, the criterion can be applied to local segments where the homogeneity assumption is appropriate.

In light of the reviewer's suggestion, we have revised our main text on page 7 line 135 and page 13 line 290-295 to include discussions on the possibility of incorporating spatial component in system contexts.

Reviewer #2 (Remarks to the Author):

While my primary expertise is in the evolutionary analysis (and modelling) of mutualism, I very much enjoyed reading this ambitious paper on ecological dynamics of mutualistic interactions. It has a big aim: to find general rules predicting population dynamics across (almost) all mutualistic systems, particularly coexistence and collapse conditions, as well as quantitative predictions (density, coexistence probability, time to cheating take-over).

In ecology and evolution, we can sometimes be too affected by what Queller called the 'tyranny of detail' (Am Nat. 2017 Apr;189(4):345-353), to even attempt finding such general rules. Abstract, higher-level (coarse-grained, as the authors call them) rules, can be very useful in guiding research, however, and I really like how the authors try to nevertheless find them.

To do so, they analyse a range of 81 different mutualisms models, based on a large set of ODEs, and aiming to reflect (most of the) diversity of mutualistic systems out there (more later on this). Deriving coexistence criteria these models, they come up with a general rule which predicts mutualism coexistence as a function of the effective benefit and stress experienced by the mutualistic partners (Eq 1). They then proceed to show how these parameters, or approximations of them, could be empirically measured and do so using a new experiment, simulations and a number of previously published (microbial) datasets.

The general rule the authors derive seems novel to me but makes a lot intuitive sense, and the application to simulations and particularly real-life datasets are both informative and convincing. Together they make for a convincing analysis, and a very useful contribution, with many potential applications in the study of mutualisms. The paper is also well written and overall a pleasure to read, and the code to replicate all theoretical analyses and simulations has all been made available.

We thank the reviewer for finding this work significant and enjoyable to read, and for making insightful and constructive suggestions and comments.

However, I have one major concern, which I feel the authors need to address because it has the potential to substantially affect the claimed generality of these results.

As I understand it, the various models analysed by the authors, all have in common that benefit is 'positively dependent on partner density' (Third line methods section, Section IIB Supplementary), although this benefit is bounded (section D Sup info, near equation II.8). However, while there are certainly situations in which this is undoubtedly true (for instance, in many of the lab microbial interactions studied empirically in this work), I am not necessarily convinced that such positive dependence on partner density hold generally among mutualistic interactions. After all, there is lots of evidence from natural systems of saturating benefits from increasing partner densities, and indeed also of shifts to negative effects beyond certain densities (e.g. Anderson and Midgley, 2007; Morris, Vázquez and Chacoff, 2010; Vannette and Hunter, 2011; Geib and Galen, 2012; Palmer and Brody, 2013 -> refs at the bottom of this review; the DeAngelis, Holland & Bronstein 2002 paper cited by

the authors also gives a few examples). Such saturation of benefits makes a lot of intuitive sense: at some points all the flowers in a plant population are pollinated, or all the herbivores chased away by protective ants. Additional ants are then at best neutral, and potentially a fitness cost (see Palmer 2013).

The authors do mention previous work on saturating benefits in their supplementary information (in the section where they discuss previous models and again when they discuss their own modelling approach), for instance discussing some of the Holland & DeAngelis work on this, but as I understand it they don't incorporate any potential saturating effects (or even shifts to negative effects at high-densities) in their models. Of course, with some exceptions, we don't generally know to what extent real-life mutualisms typically experience (bounded) positive benefits from increased partner-densities as in the current work versus when saturating or even negative density effects start to appear. However, given that we have good evidence (and theoretical reasons) to think that they may commonly exist, this could undermine the claim of generality in this paper. Consequentially, I would like to see the authors either (i) include such effects in their models and analyse their impact, or (ii) convincingly explain and show why this concern is not relevant and their conclusions would hold even under saturating benefits, or (iii) scale back the claim of generality and clearly indicate that their results may not (all) apply to mutualisms with saturating effects, and are only directly applicable to the (much?) smaller subset of mutualisms with strict positive density effects.

We thank the reviewer for the insightful comments and suggested references. We understand the concerns about generality the reviewer raised, and sincerely appreciate having our attention drawn to this apparent weakness in our presentation. We clarify here how our work does take into account the effects the reviewer has mentioned, and also describe how we have worked to make this clearer in the manuscript.

We completely agree that accounting for saturating benefits in our models is important, and they are therefore implied in all our models. As asymptotic behaviors, we modeled the effect of benefit saturating with increasing partner density. In SI section II.D. we state that our models use Hill equations to capture the effect of benefit (β) on partner fitness:

$$\frac{\beta' X_2}{X_2 + 1/\beta} \quad (\beta > 0, \beta' > 0) \quad \text{or} \quad \frac{-1}{\beta X_2 + 1} \quad (\beta > 0)$$

Fig. R5. For specific β values, the benefit that one population receives from the other can saturate.

In Fig. R5, when X_2 is not present ($X_2 = 0$) the fitness of X_1 is 0 and -1 for the left and right panel respectively. When the density of X_2 increases, the fitness of X_1 increases and then asymptotically

plateaus. We have included this figure in SI section II. D, and also mention it in SI II.C line 146-148 to clarify the importance of saturating effects.

Similarly, in Extended Data Table 1, we have included models where cost has a constant effect, or the effect increases either linearly or in a saturating fashion with partner density (refer to SI section II.D.). For the dependent cases, cost is implemented by either:

$$\frac{1}{\varepsilon X_2 + 1} \ (\varepsilon \geq 0) \text{ or } -\varepsilon X_2 \ (\varepsilon \leq 0)$$

Fig. R6. Parameter ε has a saturating or linear effect in decreasing partner's fitness.

We have included this figure in SI section II. D to provide a visualization of how ε modulates partner fitness.

We can examine the overall effects of β and ε by visualizing how our models capture the relationships between the density of X_2 and X_1 fitness. For example, model #79:

$$79. \frac{dX_1}{dt} = \frac{\beta' X_2}{X_2 + 1/\beta} X_1 (1 - X_1) - \delta (1 + \varepsilon X_2) X_1$$

captures the net effect as first saturating and then decreasing. Assuming $X_1 \ll 1$, the instant growth rate of X_1 becomes:

$$\frac{\beta' X_2}{X_2 + 1/\beta} - \delta (1 + \varepsilon X_2)$$

If we choose specific values for this function ($\beta' = 2$, $\beta = 10$, $\delta = 1$, $\varepsilon = 0.8$), we can easily see the overall fitness effect of both benefit and cost which can increase at low partner density and decrease at high partner density:

Fig. R7. Some of our models consider the case where increasing partner density leads to a biphasic fitness effect on the other population.

Using model #25, which differs from model #79 only in its constant effect of ε :

$$25. \frac{dX_1}{d\tau} = \frac{\beta'X_2}{X_2 + 1/\beta} X_1(1 - X_1) - \delta\varepsilon X_1$$

Again, assuming $X_1 \ll 1$, the instant growth rate of X_1 becomes:

$$\frac{\beta'X_2}{X_2 + 1/\beta} - \delta\varepsilon$$

If we use the same set of parameters as in Fig. R7, the fitness of X_1 is:

Fig. R8. When the effect of ε does not increase with partner density, fitness of X_1 is a monotonically increasing function of X_2 .

As the third example, a different shape can be generated using model #73:

$$73. \frac{dX_1}{d\tau} = \frac{1}{(1 + \varepsilon X_2)} \frac{\beta'X_2}{X_2 + 1/\beta} X_1(1 - X_1) - \delta X_1$$

The instant growth rate of X_1 is:

$$\frac{1}{(1 + \varepsilon X_2)} \frac{\beta'X_2}{X_2 + 1/\beta} - \delta$$

Using $\beta' = 1$, $\beta = 100$, $\delta = 1$, $\varepsilon = 5$, we get the following relationship:

Fig. R9. In this case, the overall effect of the interaction increases and decreases sharply at a low X_2 level.

Thus, with different function structures and parameter values, our model formulations capture several types of density vs. net effect curve as previously demonstrated (Morris, Vázquez and Chacoff, 2010). We have added a new section II.E to the SI; it includes Fig. R7-R9 to showcase the several types of density vs. net effect curves our models encompass.

In summary, we believe that the way we model benefit and cost is consistent with the suggestions made by the reviewer. To clarify this in the manuscript, we have edited the description of the model formulation and the rationale in the main text (page 4 line 73). As we described above, we have also edited the supplemental material section II.C, expanded SI section II.D to include Fig. R5 and Fig. R6, and added a new section II.E to include Fig. R7-R9 to showcase different types of density vs. net effect curves our models capture.

I also have some more smaller remarks:

- Please include line numbers throughout for a potential resubmission, including for the supplementary. Referencing sections of the manuscript is very cumbersome without them.

We have included line numbers on our main text and supplementary information.

- I think it would be important to highlight more explicitly in the main text that the models analysed here are all ecological models, and do not include potential evolutionary responses. It's fine to not include evolutionary dynamics, but it's important for the reader to be aware of this limitation, particularly given that eco-evolutionary dynamics could actually be important, particularly in many of the microbial mutualistic systems considered here.

We thank the reviewer for the suggestion. We now note in the main text (page 4, lines 64-65) that our models do not address evolutionary dynamics.

- Page 8 - Experimental Application of the metrics: for the first experimental application of the metric, I didn't fully understand what, if anything, the model equivalent of aTc is. The stress is imposed/modulated by IPTG, if I understand correctly, so is this just some external trigger of

Quorum-sensing, independent of stress, that for whatever reason is required in this system? Would we not want QS to be triggered by the stress/stress-inducer itself? Some clarification for the reader would be helpful here.

We apologize for the lack of clarity in our presentation. We have now included additional clarification for our first experimental system. The reviewer did understand it correctly that QS is triggered by aTc and is independent of stress, which is induced by IPTG. We have clarified this point on page 9 line 188 and 190-191 of the main text.

- Main text table: I don't very much like cancer as an example of mutualism here. I know that the authors are using a slightly wider definition of mutualism here of two populations providing reciprocal benefits (first sentence of the ms), but most typically the term is used for situations where these populations are also of different species. I would suggest instead using an interspecific protection mutualism (e.g. ant-plant).

We thank the reviewer for the suggestion. We recognize and agree that mutualism has been traditionally used to describe interspecific interactions. As general background examples, we agree that the ant-plant mutualism is more fitting than cancer. As such, we have replaced the latter with the former in Table 1.

- SI page 3 "In addition, although population collapse [...]the fitness of the benefit-receiver decreases with increasing partner density, which is contradictory to the basic logic of mutualism." -> I appreciate that given the definition of 'mutualism', at the stage where effects are negative the interaction is no longer strictly speaking a mutualism, but following my above general remark I wouldn't describe this effect as being contradictory to the basic logic of mutualism. The interaction at some density having a negative effect and then no longer being a mutualism is no way contradictory to it being a mutualist in other conditions, which I think what the models discussed here were trying to analyse.

We thank the reviewer for this suggestion. We apologize that our original description was not accurate. We have deleted this sentence.

Best wishes,
Dr. Gijbert Werner
University of Oxford
[\Signed]

References cited in this review:

Anderson, B. and Midgley, J. J. (2007) 'Density-dependent outcomes in a digestive mutualism between carnivorous *Roridula* plants and their associated hemipterans', *Oecologia*, 152(1), pp. 115–120. doi: 10.1007/s00442-006-0640-8.

Geib, J. C. and Galen, C. (2012) 'Tracing impacts of partner abundance in facultative pollination mutualisms: from individuals to populations', *Ecology*, 93(7), pp. 1581–1592. doi: 10.1890/11-1271.1.

Morris, W. F., Vázquez, D. P. and Chacoff, N. P. (2010) 'Benefit and cost curves for typical pollination mutualisms', *Ecology*, 91(5), pp. 1276–1285. doi: 10.1890/08-2278.1.

Palmer, T. M. and Brody, A. K. (2013) 'Enough is enough: the effects of symbiotic ant abundance on herbivory, growth, and reproduction in an African acacia', *Ecology*, 94(3), pp. 683–691. doi: 10.1890/12-1413.1.

Vannette, R. L. and Hunter, M. D. (2011) 'Plant defence theory re-examined: nonlinear expectations based on the costs and benefits of resource mutualisms', *Journal of Ecology*, 99(1), pp. 66–76. doi: 10.1111/j.1365-2745.2010.01755.x.

REVIEWERS' COMMENTS:

Reviewer #1 (Remarks to the Author):

The authors have put in a considerable amount of effort in addressing the reviewers' comments and I am satisfied by the changes made to the manuscript. In particular, I feel that the paper is now easier to follow, thanks to the inclusion of a specific example to introduce the general rule. Further, the analysis of the model described in (Yurtsev, Conwill and Gore, PNAS, 2016), wherein coexistence is in the form of a limit cycle rather than a fixed point, strengthens the central result of the paper and widens its applicability. I therefore recommend publication in Nature Communications.

Reviewer #2 (Remarks to the Author):

I have very much enjoyed reading the new manuscript, and the authors' very clear and extensive rebuttal. Thanks to the authors for addressing my previous concerns. I think the changes the authors made have much improved the manuscript, and made it considerably clearer. I have no further or additional comments.

Best wishes,
Dr. Gijsbert DA Werner
University of Oxford
[\Signed]

Point-by-point response

REVIEWERS' COMMENTS:

Reviewer #1 (Remarks to the Author):

The authors have put in a considerable amount of effort in addressing the reviewers' comments and I am satisfied by the changes made to the manuscript. In particular, I feel that the paper is now easier to follow, thanks to the inclusion of a specific example to introduce the general rule. Further, the analysis of the model described in (Yurtsev, Conwill and Gore, PNAS, 2016), wherein coexistence is in the form of a limit cycle rather than a fixed point, strengthens the central result of the paper and widens its applicability. I therefore recommend publication in Nature Communications.

Reviewer #2 (Remarks to the Author):

I have very much enjoyed reading the new manuscript, and the authors' very clear and extensive rebuttal. Thanks to the authors for addressing my previous concerns. I think the changes the authors made have much improved the manuscript, and made it considerably clearer. I have no further or additional comments.

Best wishes,
Dr. Gijsbert DA Werner
University of Oxford
[\Signed]

We thank both the reviewers for finding the manuscript much improved and the recommendations for publication.